# Low-methoxy pectin-containing enteral nutrition in critical care for intestinal tolerance (LOME-PECT): Study protocol for a randomized controlled trial

Shizuka Kashiwagi[1], Shunsuke Takaki[2], Yasufumi Oi[3], Hiroshi Honzawa[3], Ryo Yamamoto[4], Ikutaro Yamashita[4], Izumi Ohki[5], Shigeki Fujitani[6], Akiyoshi Nagatomi[6], Yuki Ohshima[7], Minoru Yoshida[8], Hideki Yoshida[8], Miyuki Kurisu[8], Yuji Takahashi[9], Hideki Hashimoto[9], Yasuaki Koyama[9], Junji Hatakeyama[10], Satoru Shinoda[11], Nobuyuki Yokoyama[2], Kensuke Nakamura [2]*

1 Department of Critical Care Medicine, Yokohama City University Medical Center, Yokohama, Japan, 2 Department of Anesthesiology and Critical Care Medicine, Yokohama City University School of Medicine, Yokohama, Japan, 3 Department of Emergency Medicine, Yokohama City University School of Medicine, Yokohama, Japan, 4 Department of Emergency and Critical Care Medicine, Keio University School of Medicine, Tokyo, Japan, 5 Catering and Nutrition Office, Keio University Hospital, Tokyo, Japan, 6 Department of Emergency and Critical Care Medicine, St.Marianna University School of Medicine, Kawasaki, Japan, 7 Division of Clinical Nutrition, St. Marianna University School of Medicine, Kawasaki, Japan, 8 Department of Emergency and Critical Care Medicine, St. Marianna University School of Medicine, Yokohama City Seibu Hospital, Yokohama, Japan, 9 Department of Emergency and Critical Care Medicine, Hitachi General Hospital, Hitachi, Japan, 10 Department of Emergency and Critical Care Medicine, Osaka Medical and Pharmaceutical University, Takatsuki, Japan, 11 Department of Biostatistics, School of Medicine, Yokohama City University, Yokohama, Japan

* mamashockpapashock@yahoo.co.jp

## Abstract

### Background

Enteral nutrition is preferable over parenteral nutrition for critically ill patients, but is often discontinued due to enteral feeding intolerance. Diarrhea is one of the most common causes of the discontinuation of enteral nutrition and may be attributed to the composition of enteral formulas. Dietary fiber attenuates diarrhea by normalizing the intestinal microbiota, providing energy for colonic epithelial cells, and exerting a thickening effect on intestinal contents. We herein conducted a randomized controlled trial (RCT) to test the hypothesis that the administration of an enteral formula containing low-methoxy pectin, a type of dietary fiber, more effectively ameliorates diarrhea in critically ill adult patients than a similar composition without pectin.

### Methods

A protocol for planning a multicenter, parallel-group, open-label RCT is described herein. Enrolled patients are those ≥18 years of age with the indication of enteral nutrition by gastric access. Overall, 200 patients will be randomized into an

**Data availability statement:** No datasets were generated or analysed during the current study. All relevant data from this study will be made available upon study completion.

**Funding:** This trial is supported by grants from Otsuka Pharmaceutical Co., Ltd. (No. MF23-2). There was no additional external funding received for this study

**Competing interests:** This trial is supported by grants from Otsuka Pharmaceutical Co., Ltd. (No. MF23-2). There was no additional external funding received for this study." There are no other competing interests to be reported for all authors.

**Abbreviations:** EN, enteral nutrition; PN, parenteral nutrition; ICU, intensive care unit; RR, relative risk; CI, confidence interval; RCT, randomized controlled trial; EDC, electronic data capture; FAS, full analysis set; SAS, safety analysis set, SOFA, Sequential Organ Failure Assessment; MUST, Malnutrition Screening Tool.

intervention group administered an enteral formula containing low-methoxy pectin and a control group administered an enteral formula with similar components, but without pectin at a ratio of 1:1. Each enteral formula will be administered for 3 days or longer. There are no restrictions on other treatments. The primary outcome is the incidence of diarrhea as defined by Bristol Scale 5, 6, or 7. Secondary outcomes include the rate of EN failure, the survival rate, the lengths of ICU and hospital stays, and nutritional endpoints.

## Discussion

The present study examines the effects of a low-methoxy pectin-containing enteral formula on enteral feeding intolerance, including diarrhea, in critically ill patients. The results obtained may provide new considerations regarding the selection of enteral formulas for critically ill patients.

## Trial registration

jRCTs031230684 registered on 08 Mar 2024, https://jrct.niph.go.jp/en-latest-detail/jRCTs031230684.

## Introduction

### Background and rationale

Enteral nutrition (EN) may be preferable to parenteral nutrition (PN) as nutrition therapy for critically ill patients. The initiation of EN 24−48 hours after admission to the intensive care unit (ICU) has been shown to reduce infectious complications [1–3]. Based on these findings, international guidelines recommend early EN [4]. However, EN is often discontinued due to enteral feeding intolerance, including a large gastric residual volume, nausea, vomiting, constipation, and diarrhea [5]. Diarrhea occurs in 37.7–73.8% of critically ill patients and is one of the most common causes of the discontinuation of EN [6]. Therefore, the prevention and management of diarrhea are regarded as important issues for accomplishing nutrition therapy. The causes of diarrhea in critically ill patients vary widely and include the administration of broad spectrum antimicrobial agents, intestinal ischemia, and *Clostridium difficile* infection. If these causes are absent, diarrhea may be related to EN itself, including the composition of an enteral formula and the method of administration. A recent prospective cohort study identified EN (relative risk (RR) 1.23, 95% confidence interval (CI) 1.16-1.31) as the largest risk factor for diarrhea in ICU patients, followed by the duration of antimicrobial therapy (RR 1.02, 95% CI 1.02-1.03) and suppository use (RR 1.14, 95% CI 1.06-1.22) [6]. Several characteristics of enteral formulas have been associated with diarrhea, including high osmolarity, the amount of carbohydrates, fat, protein-based formulas, and no dietary fiber content.

Dietary fiber is an indigestible carbohydrate, and its fermentation by intestinal bacteria yields short-chain fatty acids, which provide energy for intestinal bacteria and

colonic epithelial cells and also exert a thickening effect on intestinal contents, thereby attenuating diarrhea [7]. Pectin, a type of soluble dietary fiber, gelatinizes in an acidic environment, thereby delaying the arrival of gastric contents to the small intestine. In addition, a recent study using a rat model showed that low methoxy pectin, the degree of esterification of which is less than 50%, was particularly effective at regulating stool properties and suppressing diarrhea [8]. In our previous retrospective study, enteral formulas containing low methoxy pectin ameliorated diarrhea in ICU patients [9]. We herein hypothesized that the use of a low methoxy pectin-containing enteral formula suppresses diarrhea in critically ill patients, even that with high lipid and protein contents. We will conduct a multicenter randomized controlled trial (RCT) in which patients admitted to the ICU and indicated for EN are randomized into a low methoxy pectin-containing enteral formula group or control group.

## Trial registration

Trial identifier: jRCTs031230684. Registered on 08 Mar 2024.

Registry name: Low-Methoxy Pectin containing Enteral nutrition in Critically care for intestinal Tolerance (LOME-PECT trial)

## Protocol version

Ver. 2.0, updated on 18 June 2024. Copies of the protocol are shown in S2 File (English) and S3 File (Japanese).

## Name and contact information of the trial sponsor

Medical Foods Research Institute, OS-1 Division, Otsuka Pharmaceutical Factory, Inc.

115 Kuguhara, Tateiwa, Muya-cho, Naruto, Tokushima 772–8601, Japan

## Role of the sponsor

The sponsor provides enteral formulas, but has no authority regarding the implementation of this study or data collection, its management, analysis, or interpretation. The joint clinical research agreement document is shown in S1 File.

## Objectives

The primary aim of this study is to examine the efficacy of an enteral formula containing low methoxy pectin against diarrhea in adult critically ill patients.

## Trial design

The LOME-PECT trial is a multicenter, parallel-group, open-label RCT. The study is Specified Clinical Research under the Clinical Trials Act in Japan. This clinical study will enroll 200 adult patients admitted to the ICU. The study period is between March 08th, 2024 and March 31st, 2028. The recruitment started on 26 June 2024 and will continue until 31 March 2025.

## Methods

### Study setting

The present study is conducted at 6 ICUs in Japan as listed below.

Yokohama City University Hospital, Yokohama

Keio University Hospital, Tokyo

St. Marianna University Hospital, Kawasaki

St. Marianna University School of Medicine, Yokohama City Seibu Hospital, Yokohama

Hitachi General Hospital, Hitachi
Osaka Medical and Pharmaceutical University Hospital, Takatsuki

**Eligibility criteria**

 **Inclusion criteria.** We include the following patients:

- Admitted to the ICU (any reason for admission)

- Age ≥ 18 years

- Indication for EN by gastric feeding

- Agreeable to provide written informed consent for study participation.

 **Exclusion criteria.** We exclude the following patients:

- Received enteral feeding within 30 days prior to enrollment

- Presenting with diarrhea (Bristol scale ≥5) at enrollment

- Contraindications or medical inappropriateness (including allergies) to gastric feeding or enteral formulas

- "Do Not Resuscitate" policy

- Received EN via gastrostoma or percutaneous trans-esophageal gastro-tubing

- Judged as inappropriate by a physician

**Who will take informed consent?**

The principal investigator or sub-investigator at each site will obtain written consent for participation after confirming the eligibility of patients.

**Additional consent provisions for the collection and use of participant data and biological specimens in ancillary studies**

Plasma, whole blood, and fecal samples will be optionally collected on Day 0 (before the initiation of EN), Day 3, and Day 7 for patients who have given their consent, and will be stored at −80°C. Samples will be anonymized at each institution and stored for a secondary analysis.

**Interventions**

 **Explanation for the choice of comparators.** Patients in the control group will receive GLUCERNA® REX as the enteral formula. GLUCERNA® REX is a pectin-free enteral formula with a similar nutrient density and macronutrient balance to the test formula, HINEX® RENUTE.

 **Interventions for each group with sufficient detail to allow replication, including how and when they will be administered.** The SPIRIT Figure is shown in Fig 1 and the study outline is in Fig 2. After obtaining written consent from potentially eligible patients, the necessary tests will be conducted to confirm eligibility. The use of test data obtained before consent will be allowed. EN will be started at the discretion of the physician in charge according to medical indications. There is no time limit between enrollment and the initiation of EN, however, there are minimal practical barriers to starting EN in the enrolled patients, and efforts to align enrollment timing with EN initiation are encouraged in order to minimize cases where EN could not be commenced post-enrollment. Therefore, EN is expected to be initiated within 24 hours.

| TIMEPOINT | Enrolment<br>ICU admission ~ Day 0 | Allocation<br>Day 0 | Post-allocation | | | | | Close-out<br>Day 28 | Discontinuation |
|---|---|---|---|---|---|---|---|---|---|
| | | | Day 1 | Day 2 | Day 3 | Day 7 | Day 14 | | |
| Permissible range (days) | | 0 | 0 | 0 | 0 | ± 2 | ± 2 | +7 | ± 3 |
| **Enrolment:** | | | | | | | | | |
| Eligibility screen | ✓ | | | | | | | | |
| Informed consent | ✓ | | | | | | | | |
| Allocation | | ✓ | | | | | | | |
| **Interventions:** | | | | | | | | | |
| GLUCERNA® REX administration [a] | | | ←————————→ | | | | | | |
| HINEX® RENUTE administration [a] | | | ←————————→ | | | | | | |
| **Assessments:** | | | | | | | | | |
| Gastrointestinal symptoms | | ✓ | ✓ | ✓ | ✓ | | | | |
| State of defecation [b] | | ✓ | ✓ | ✓ | ✓ | | | | |
| Confirmation of laxative administration | | ✓ | ✓ | ✓ | ✓ | | | | |
| Blood sampling [c] | | ✓ | | | ✓ | ✓ | | | |
| Fecal sampling [c] | | ✓ | | | ✓ | ✓ | | | |
| Blood test | ✓ | ✓ | ✓ | | ✓ | ✓ | ✓ | | ✓ |
| Confirmation of survival | | | | | | | | ✓ | |
| Observation of adverse events | | ←————————————————→ | | | | | | | |

**Fig 1. The SPIRIT figure.** Schedule of enrolment, interventions, and assessments. a: Dosing after Day 3 is allowed under clinical judgment. b: Only patients for whom additional consent was obtained. c: Bristol Scale ≥5, 200 g/day, or 300 ml/day.

We adopt HINEX® RENUTE as the enteral formula in the intervention group because it contains low-methoxy pectin with an optimal composition for nutrition therapy for critically ill patients: low carbohydrate and high protein contents. In the control group, GLUCERNA® REX is used as the enteral formula with a similar lipid content and energy density (kcal/ml) to HINEX® RENUTE in terms of aligning the composition affecting diarrhea. HINEX® REUNUTE is a liquid nutrition formula that contains energy density 1.0 kcal/mL, collagen and soy peptide as the nitrogen source, osmotic pressure 380 mOsm/L, and a protein, fat, and carbohydrate balance of 24, 50, and 26%, respectively. It contains 0.68 g pectin per 100 kcal. GLUCERNA® REX is a liquid nutrition formula that contains energy density 1.0 kcal/mL, soy protein, osmotic pressure 560 mOsm/L, and a protein, fat, and carbohydrate balance of 17, 50, and 33%, respectively. It contains 0.9 g dietary fiber per 100 kcal. The treatment group will receive HINEX® RENUTE and the control group will receive GLUCERNA® REX by continuous gastric administration via a feeding tube for 3 days. At the physician's discretion, any enteral formula may be administered until Day 7, except GLUCERNA® REX, in the treatment group and HINEX® RENUTE in the control group. After Day 8, neither the test nor control formula will be administered in both groups. The dose rate will start at 10 ml/h. If there are no adverse events, the dose rate will be increased by 10 ml/h every 8 hours. The target energy dose is not specified and may be selected by the physician, but will not exceed 20 kcal/kg until Day 3.

## Criteria for discontinuing or modifying allocated interventions

If a participant does not meet eligibility after enrollment and prior to administration of the study nutrient, the intervention will be terminated and no study nutrient will be administered.

## Strategies to improve adherence to interventions

Not applicable in the present study.

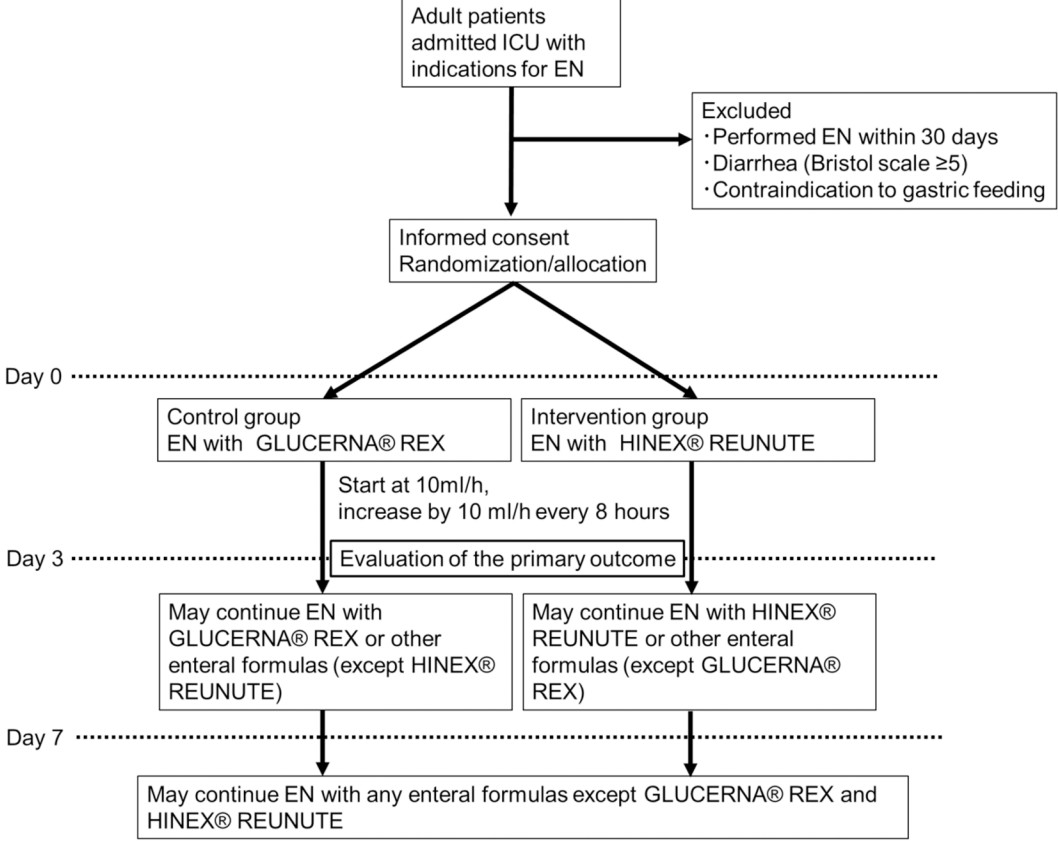

**Fig 2. The study outline.**

### Relevant concomitant care permitted or prohibited treatments during the trial

Enteral formulas other than the study nutrients will not be used in either group until the primary outcome is evaluated. However, non-nutritional medications may be administered, and PN may be used as appropriate. There are no restrictions on treatments other than nutrition therapy.

### Outcomes

**Primary outcome.** The incidence of diarrhea (as defined by Bristol Scale 5, 6, or 7) within 3 days of the administration of study enteral formulas is evaluated as the primary outcome. Since the residence time of enteral formulas in the intestinal tract is approximately 24 hours and diarrhea, if accompanied, is expected to occur within the following 24 hours, it is considered sufficient to detect the onset of diarrhea within 3 days of starting EN. The Bristol Scale is applied as a measure of diarrhea, and a score of 5–7 is defined as diarrhea based on previous studies and a recent review [9–11].

**Secondary outcomes.** Secondary outcomes are (1) the Bristol Scale, the presence or absence of diarrhea (≥200 g/day or 300 ml/day) and watery diarrhea (Bristol Scale 7) on Days 1, 2, and 3; (2) the occurrence of diarrhea (defined by Bristol Scale 5–7) in the first week; (3) the rate of EN failure on Days 3 and 7; (4) the duration of EN; (5) the daily doses of energy (kcal/day) and protein (g/day) for EN and PN during the first 7 days; (6) the survival rate on Day 28; (7) the length of ICU stay; (8) the length of hospital stay; (9) ventilator days; (10) nutritional endpoints: white blood cells, total

lymphocyte count, C reactive protein, albumin, transthyretin, cholesterol (total, low density lipoprotein, and high density lipoprotein) and triglycerides on Days 7 and 14; (11) the Barthel index on Day 28; (12) the occurrence of adverse events, including enteral feeding intolerance and infectious complications.

## Participant timeline

The participant timeline is shown in Fig 2.

## Sample size

Based on our previous retrospective study, we estimated the incidence of diarrhea in this study [9]. The incidence of diarrhea in the control group using GLUCERNA® REX was estimated to be 50%, while that in the treatment group using HINEX® RENUTE was 30%. By applying Pearson's chi-square test to the predicted incidence of diarrhea at a two-sided significance level of 5%, the number of cases required to maintain 80% power was calculated to be 93 in each group. In consideration of some dropouts, the target number of cases was set at 100 in each group, for a total of 200 cases.

## Recruitment

This study recruits patients for whom EN is indicated regardless of the reason for ICU admission; therefore, achieving the required number of participants is feasible.

## Assignment of interventions: allocation

**Sequence generation and implementation.**  Block randomization is performed using electronic data capture (EDC). The EDC site automatically generates a random sequence in each hospital; therefore, stratification is performed only for the facilities. Study physicians include participants on the EDC site, on which their allocation and assigned procedure are rapidly noted.

## Concealment mechanism

Researchers at one hospital are blinded to the assignments or outcomes of patients at other hospitals.

## Assignment of interventions: Blinding

**Who will be blinded.**  Difficulties are associated with blinding enteral formulas in this study due to hygiene concerns and potential clinical interference. Therefore, we planned this study as a non-blinded RCT. Trial participants, care providers, outcome assessors, and data analysts are not blinded. However, two or more ICU nurses who are not involved in this study assess diarrhea using the Bristol Scale, which they are familiar with in their clinical practice, and the secondary outcome, the amount of diarrhea, is an objective measure. Therefore, observer bias may be minimized.

## Data collection and management

**Plans for the assessment and collection of outcomes.**  The outcomes and baselines of all participants are collected and assessed on EDC. All data collection forms were newly created for use only in this study by TXP Medical Corp. Japan.

Baseline data to be collected consist of background characteristics, disease category, Sequential Organ Failure Assessment (SOFA), nutrition screening based on the Malnutrition Screening Tool (MUST), laboratory test, and gastrointestinal symptoms including Bristol scale [11–13]. Details of these items are shown in S1 Table. Descriptive statistics will be calculated to summarize these variables. Between-group inferential statistical comparisons will not be conducted, as these analyses are intended for descriptive purpose only.

### Plans to promote participant retention and complete follow-ups

Participant retention and follow-ups are expected to be easy in this study because patients are observed only during their hospitalization in the ICU.

### Data management

Patient data are stored in raw medical records at each hospital and anonymized EDC for at least five years. Changes in EDC are preserved with a log showing the information of who and when they are changed.

### Confidentiality

All patient data are anonymized in the EDC system. Only study physicians, who were given the original ID and password, access EDC and solely input data on patients at their facility. The statistician and central monitor have exclusive access to all participants' data.

### Plans for collection, laboratory evaluations, and the storage of biological specimens for genetic and molecular analyses in this trial/future use

If we obtain additional consent from each patient, we will collect plasma, whole blood, and fecal samples on Days 0, 3, and 7 and store them at −80°C. These samples will be used to perform additional analyses on lipid and protein metabolism and to examine the effects of diarrhea on nutrient metabolism in severe disease.

### Statistical methods

**Statistical methods for primary and secondary outcomes.** Statistical analyses are performed using an intention-to-treat analysis with a full analysis set (FAS) and safety analysis set (SAS). FAS is defined as all subjects without violations of the main eligibility criteria (selection and exclusion criteria) or conflicts with discontinuation and dropout criteria. SAS is defined as all subjects who received the study treatment. The efficacy analysis is performed with FAS. The safety analysis will be performed with SAS. No statistical methods are used to handle missing data (e.g., multiple imputation).

Frequencies will be calculated for binary data, including the incidence of diarrhea, the rate of EN failure, and the survival rate on Day 28. Continuous variables assumed to follow a normal distribution, such as age, height, and weight, will be presented as means and standard deviations. Variables not assumed to follow a normal distribution, such as ICU stay, hospital stay, and ventilator days, will be reported as medians and interquartile ranges. Pearson's chi-square test will be employed to compare groups for the primary endpoint. In principle, a two-tailed test is conducted with a significance level of 5% and CI of 95%. To examine the interaction effect, subgroup analyses will be performed with the following subgroups: sex, age (<70 years old/ ≥ 70 years old), BMI (≤20/ 20 < BMI ≤ 30/ ≥ 30), SOFA scores (less than 2 points in one or more items/others) [12], and disease category (infectious disease, heart failure, respiratory failure, stroke, renal or metabolic diseases, postoperative, post-cardioplumonary resuscitation, trauma, other).

Regarding secondary endpoints and adverse events, between-group comparisons for binary outcomes will be analyzed using confidence intervals for differences in population proportions, while continuous outcomes will be analyzed by confidence intervals in means or medians. Additionally, as a post-hoc analysis, a subgroup analysis will be conducted based on the above results.

### Oversight and monitoring

**Composition of the coordinating center and trial steering committee.** The principal investigator and study coordinator is Kensuke Nakamura, Yokohama City University Hospital. The data manager is Shizuka Kashiwagi, Yokohama City University Hospital. The statistical analysis manager is Satoru Shinoda, Yokohama City University.

Nobuyuki Yokoyama, Yokohama City University Hospital plays a role in coordination and study management. The Certification of Clinical Trials Review Board is established in Yokohama City University as the trial steering committee.

**Composition of the data monitoring committee, its role, and its reporting structure.** The monitoring manager is Shizuka Kashiwagi, Yokohama City University Hospital. Central monitoring is performed based on CRF data collected at the data center. The monitoring manager will conduct central monitoring twice a year at the request of the principal investigator. On-site monitoring is performed at each hospital by monitors appointed by the monitoring committee.

### Interim analyses

No interim analysis is performed in this study

### Adverse event reporting and harm

As with general EN therapy, gastrointestinal complications, such as diarrhea, nausea/vomiting, abdominal pain, intestinal ischemia, ileus, and hemorrhage, are expected in this study. Infectious complications, including pneumonia and bacteremia, are also evaluated in this study. Adverse events need to be reported on medical records and EDC, with causal associations with the intervention, dates, severity, with or without any treatments, and outcomes. Severity is evaluated according to Common Terminology Criteria for Adverse Event v5.0. Pneumonia is defined as an infiltrative shadow on a chest radiograph and two of the following: >38°C fever, a white blood cell count <4000 or >11000/μL, and purulent sputum.

### Frequency and plans for auditing trial conduct

No audits will be conducted in this study

### Plans for communicating important protocol amendments to relevant parties (e.g., trial participants and Ethics Committees)

If the study protocol is modified, it has to be approved by the Certified Review Board at Yokohama City University and patients will provide written informed consent for amendments.

### Provisions for post-trial care

Physicians need to perform rapid and appropriate treatments if any diseases or complications occur. This study is affiliated with clinical trial insurance, and death or damage related to this study are compensated through this insurance for all participants. Additional healthcare costs are covered by the national health insurance system.

### Dissemination plans

The results of this study will be presented at academic conferences and will also be published in scientific journals.

### Plans to provide access to the full protocol, participant-level data, and statistical codes

This study protocol is registered with the registration number jRCTs031230684 on the Japan Registry of Clinical Trials.

### Ethics approval and consent to participate

The study protocol was approved by the centralized IRB at The Yokohama-City University (Yokohama, Japan; #CRB23−016), and subsequently, the approved protocol was approved at the IRB at each collaborating center. Written informed consent will be sought from all potential participants or their representatives. The study was designed and will be implemented according to the guidelines in the Helsinki Declaration. The current protocol of the LOME-PECT trial is reported according to the SPIRIT reporting guidelines (S4 File) [14].

## Discussion

We are conducting the present study to test the hypothesis that a low methoxy pectin-containing enteral formula attenuates diarrhea more than conventional enteral formulas in critically ill adult patients. EN has advantages over PN, which include the preservation of intestinal immunity, the prevention of bacterial translocation, and a lower cost. Nevertheless, recent large RCTs failed to demonstrate the superiority of EN over PN [15,16], and guidelines from the American Society for Parenteral and Enteral Nutrition recommend that EN or PN is acceptable for initial nutrition therapy [17]. One of the reasons for the disadvantages of EN is enteral feeding intolerance, including diarrhea. Diarrhea is associated with hemodynamic instability and electrolyte abnormalities, as well as impaired intestinal perfusion and intestinal immunity, which may delay discharge from the ICU and hospital [10,18,19]. In addition, malnutrition and immune disorders are associated with the development of post-intensive care syndrome and persistent inflammation, immunosuppression, and catabolism syndrome and may have a negative impact on long-term prognosis [20,21]. Although the causes of diarrhea vary, EN itself may often be the cause in critically ill patients. A previous study reported that the composition of an enteral formula may contribute to the amelioration of diarrhea, including low osmolarity, low lipid, peptide-based nutrition as a nitrogen source, and the inclusion or addition of dietary fiber [18]. Recent meta-analyses showed that dietary fiber supplementation reduced enteral feeding intolerance, including diarrhea, constipation, and vomiting [22,23]. However, although various types of dietary fiber are commercially available (e.g., polysaccharide, psyllium, partially hydrolyzed guar gum, and pectin), evidence for which dietary fiber is effective is limited [23]. Pectin is a dietary fiber whose viscosity increases by cross-linking in acidic environments. Recently developed low-methoxy pectin presents higher viscosity in acidic environments than conventional pectin and has ameliorated diarrhea in animal experiments [8]. Dietary fiber supplementation on EN, known as prebiotics, is expected to maintain intestinal immune function and reduce intestinal-derived inflammation, particularly in combination with probiotics. Nevertheless, limited information is available on dietary fiber in the guidelines for nutritional therapy for critically ill patients [24,25]. Several RCTs have examined and failed to show the effects of regular pectin [26,27]. However, to the best of our knowledge, this is the first RCT to investigate the effects of an enteral formula containing low methoxy pectin on diarrhea control. The results of this trial will provide new information for the management of enteral feeding intolerance and for the selection of enteral formulas for ICU patients. Furthermore, it may provide us with strategies to maximize the superiority and efficacy of EN over PN.

## Trial status

The current version of the protocol is 2.0, which was updated on 18 June 2024. The recruitment started on 26 June 2024.

## Supporting information

**S1 Table. List of baseline data to be collected.**
(DOCX)

**S1 File. Joint clinical research agreement.**
(DOCX)

**S2 File. Copy of protocol (English).**
(DOCX)

**S3 File. Copy of protocol (Japanese).**
(DOCX)

**S4 File. SPIRIT checklist.**
(DOC)

# Acknowledgments

Not applicable

# Author contributions

**Conceptualization:** Shizuka Kashiwagi, Nobuyuki Yokoyama, Kensuke Nakamura.

**Formal analysis:** Satoru Shinoda, Nobuyuki Yokoyama.

**Funding acquisition:** Kensuke Nakamura.

**Investigation:** Shunsuke Takaki, Yasufumi Oi, Hiroshi Honzawa, Ryo Yamamoto, Ikutaro Yamashita, Izumi Ohki, Shigeki Fujitani, Akiyoshi Nagatomi, Yuki Ohshima, Minoru Yoshida, Hideki Yoshida, Miyuki Kurisu, Yuji Takahashi, Hideki Hashimoto, Yasuaki Koyama, Nobuyuki Yokoyama.

**Project administration:** Kensuke Nakamura.

**Supervision:** Kensuke Nakamura.

**Writing – original draft:** Shizuka Kashiwagi, Satoru Shinoda.

**Writing – review & editing:** Shunsuke Takaki, Yasufumi Oi, Hiroshi Honzawa, Ryo Yamamoto, Ikutaro Yamashita, Izumi Ohki, Shigeki Fujitani, Akiyoshi Nagatomi, Yuki Ohshima, Minoru Yoshida, Hideki Yoshida, Miyuki Kurisu, Yuji Takahashi, Hideki Hashimoto, Yasuaki Koyama, Nobuyuki Yokoyama, Junji Hatakeyama, Kensuke Nakamura.

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
