## [Decision Letter · Decision Letter 0]

22 Apr 2025

Dear Dr. Nakamura,

Thank you for submitting your manuscript to PLOS ONE. After careful consideration, we feel that it has merit but does not fully meet PLOS ONE’s publication criteria as it currently stands. Therefore, we invite you to submit a revised version of the manuscript that addresses the points raised during the review process.

We look forward to receiving your revised manuscript.

Kind regards,

Belgin Sever, Ph.D.

Academic Editor

PLOS ONE

Journal Requirements:

“This trial is supported by grants from Otsuka Pharmaceutical Co., Ltd. (No. MF23-2)”

“The LOME-PECT trial is supported by grants from Otsuka Pharmaceutical Co., Ltd. (No. MF23-2)”

“This trial is supported by grants from Otsuka Pharmaceutical Co., Ltd. (No. MF23-2)”

Additional Editor Comments (if provided):

Reviewers' comments:

Reviewer's Responses to Questions

**Comments to the Author**

1. Does the manuscript provide a valid rationale for the proposed study, with clearly identified and justified research questions?

Reviewer #1: Yes

2. Is the protocol technically sound and planned in a manner that will lead to a meaningful outcome and allow testing the stated hypotheses?

Reviewer #1: Yes

3. Is the methodology feasible and described in sufficient detail to allow the work to be replicable?

Reviewer #1: Yes

4. Have the authors described where all data underlying the findings will be made available when the study is complete?

Reviewer #1: Yes

5. Is the manuscript presented in an intelligible fashion and written in standard English?

Reviewer #1: Yes

You may also provide optional suggestions and comments to authors that they might find helpful in planning their study.

Reviewer #1: The authors present a protocol for a randomized control trial comparing to enteric nutrition (EN) formulas in critically ill patients in the ICU requiring EN in Japan. The manuscript will be strengthened if the authors consider the following points.

1. Line 305-306: authors state there is no time limit between enrollment and initiation of EN, but is there a typical time window in which they expect this to be started? It might be helpful to provide a typical range of time.

2. Will any comparisons of the groups be made to assess how well randomization balanced the groups? It would be important to know if there are any differences at "baseline".

3. line 453: authors refer to subgroup analyses, but they include some continuous variables in the list of variables that will be considered for sub-groups. The protocol (in supplemental material) includes more specific sub-groups for age, BMI, and SOFA, which would be helpful to include in the manuscript. SOFA should also be defined. The protocol also includes subcategories of diseases for the sub-groups, so that should also be included in the manuscript.

4. Further details should be provided for the analysis of the secondary outcomes. There is no mention of how comparisons between groups will be made and there are different types of outcomes included as secondary outcomes than the primary outcome, so analytic techniques will presumably be different.

5. Lines 570-571: authors state that written informed consent will be obtained from all potential participants - is this really feasible for someone in the ICU?

**Do you want your identity to be public for this peer review?** For information about this choice, including consent withdrawal, please see our Privacy Policy

Reviewer #1: No

---

## [Author Response · Author response to Decision Letter 1]

16 May 2025

Responses to the Reviewers

Reviewer(s)' Comments to the Author:

Journal Requirements:

1. Please ensure that your manuscript meets PLOS ONE's style requirements, including those for file naming. The PLOS ONE style templates can be found at https://journals.plos.org/plosone/s/file?id=wjVg/PLOSOne_formatting_sample_main_body.pdf and https://journals.plos.org/plosone/s/file?id=ba62/PLOSOne_formatting_sample_title_authors_affiliations.pdf.

Our response: According to the style requirements, we changed the font size and file name and moved the figure caption.

“This trial is supported by grants from Otsuka Pharmaceutical Co., Ltd. (No. MF23-2)”

Our response: We provided an amended Funding Statement. We also added in the cover letter that we received no additional external funding for this study.

3. Thank you for stating the following in the Acknowledgments Section of your manuscript: “The LOME-PECT trial is supported by grants from Otsuka Pharmaceutical Co., Ltd. (No. MF23-2)” We note that you have provided additional information within the Acknowledgements Section that is not currently declared in your Funding Statement. Please note that funding information should not appear in the Acknowledgments section or other areas of your manuscript. We will only publish funding information present in the Funding Statement section of the online submission form. Please remove any funding-related text from the manuscript and let us know how you would like to update your Funding Statement. Currently, your Funding Statement reads as follows:

“This trial is supported by grants from Otsuka Pharmaceutical Co., Ltd. (No. MF23-2)” Please include your amended statements within your cover letter; we will change the online submission form on your behalf.

Our response: In accordance with the journal’s requirement, we have removed the section entitled ”Competing Interests” and “Source of Funding” from the manuscript to exclude funding sources in the manuscript. We have also amended funding statement in the cover letter.

Our response: We added supporting information section to the manuscript and added supporting files including SPIRIT checklist.

Our response: We have verified that the references are complete and correct.

Reviewer #1:

Reviewer #1: The authors present a protocol for a randomized control trial comparing to enteric nutrition (EN) formulas in critically ill patients in the ICU requiring EN in Japan. The manuscript will be strengthened if the authors consider the following points.

Our response: We wish to express our appreciation to the reviewer for careful review and insightful comments on our paper.

1. Line 305-306: authors state there is no time limit between enrollment and initiation of EN, but is there a typical time window in which they expect this to be started? It might be helpful to provide a typical range of time.

Our response: We appreciate the reviewer’s insightful comment. As noted, there is no strict time limit between enrollment and initiation of EN. However, in practice, EN is typically initiated within 24 hours following enrollment. Moreover, in order to minimize cases where EN could not be commenced post-enrollment, we take care to initiate EN immediately after the enrollment. We have clarified these points in the revised manuscript as follows.

Lines: 308-312

There is no time limit between enrollment and the initiation of EN. However, there are minimal practical barriers to starting EN in the enrolled patients, and efforts to align enrollment timing with EN initiation are encouraged in order to minimize cases where EN could not be commenced post-enrollment. Therefore, EN is expected to be initiated within 24 hours.

2. Will any comparisons of the groups be made to assess how well randomization balanced the groups? It would be important to know if there are any differences at "baseline".

Our response: We appreciate the reviewer’s insightful comment. We will collect background factors of enrolled patients and calculate summary statistics. However, between-group comparisons will not be conducted, as the aim is to characterize the overall study sample rather than to test for group differences. To clarify this point, we have revised the manuscript and added a supplemental table for background data to be collected. Accordingly, we have added abbreviations.

Lines 431-437

Baseline data to be collected consist of background characteristics, disease category, Sequential Organ Failure Assessment (SOFA), nutrition screening based on the Malnutrition Screening Tool (MUST), laboratory test, and gastrointestinal symptoms including Bristol scale. Details of these items are shown in S1Table. Descriptive statistics will be calculated to summarize these variables. Between-group inferential statistical comparisons will not be conducted, as these analyses are intended for descriptive purpose only.

3. line 453: authors refer to subgroup analyses, but they include some continuous variables in the list of variables that will be considered for sub-groups. The protocol (in supplemental material) includes more specific sub-groups for age, BMI, and SOFA, which would be helpful to include in the manuscript. SOFA should also be defined. The protocol also includes subcategories of diseases for the sub-groups, so that should also be included in the manuscript.

Our response: We appreciated the Reviewer’s comment on this point. We revised the details of subgroup analyses as follows. We also cited an article which refers to SOFA.

Lines: 477-482

To examine the interaction effect, subgroup analyses will be performed with the following subgroups: sex, age (<70 years old/ ≥70 years old), BMI (≤20/ 20< BMI ≤30/ ≥30), SOFA scores (less than 2 points in one or more items/others), and disease category (infectious disease, heart failure, respiratory failure, stroke, renal or metabolic diseases, postoperative, post-cardioplumonary resuscitation, trauma, other).

4. Further details should be provided for the analysis of the secondary outcomes. There is no mention of how comparisons between groups will be made and there are different types of outcomes included as secondary outcomes than the primary outcome, so analytic techniques will presumably be different.

Our response: We appreciate the reviewer’s insightful comment. We revised the manuscript as follows for the secondary outcomes analysis method.

Lines 475-497

Frequencies will be calculated for binary data, including the incidence of diarrhea, the rate of EN failure, and the survival rate on Day 28. Continuous variables assumed to follow a normal distribution, such as age, height, and weight, will be presented as means and standard deviations. Variables not assumed to follow a normal distribution, such as ICU stay, hospital stay, and ventilator days, will be reported as medians and interquartile ranges.

Pearson’s chi-square test will be employed to compare groups for the primary endpoint. In principle, a two-tailed test is conducted with a significance level of 5% and CI of 95%. To examine the interaction effect, subgroup analyses will be performed with the following subgroups: sex, age (<70 years old/ ≥70 years old), BMI (≤20/ 20< BMI ≤30/ ≥30), SOFA scores (less than 2 points in one or more items/others)(12), and disease category (infectious disease, heart failure, respiratory failure, stroke, renal or metabolic diseases, postoperative, post-cardioplumonary resuscitation, trauma, other).

Regarding secondary endpoints and adverse events, between-group comparisons for binary outcomes will be analyzed using confidence intervals for differences in population proportions, while continuous outcomes will be analyzed by confidence intervals in means or medians. Additionally, as a post-hoc analysis, a subgroup analysis will be conducted based on the above results.

5. Lines 570-571: authors state that written informed consent will be obtained from all potential participants - is this really feasible for someone in the ICU?

Our response: We appreciated the Reviewer’s comment. As the reviewer pointed out, because this study involves ICU patients, it is often not possible to obtain consent directly from participants. In such cases, written informed consent will be sought from the participant’s legally authorized representatives including parents, siblings, or children. Additionally, due to the nature of ICU admissions, particularly during holidays or nighttime hours, it may not be feasible to obtain consent from all potential participants immediately. We revised the manuscript to clarify this point and ensure consistency

Lines 612-614

Written informed consent will be sought from all potential participants or their representatives.

---

## [Decision Letter · Decision Letter 1]

3 June 2025

Low-methoxy pectin-containing enteral nutrition in critical care for intestinal tolerance (LOME-PECT): study protocol for a randomized controlled trial.

PONE-D-24-55750R1

Dear Dr. Nakamura,

We’re pleased to inform you that your manuscript has been judged scientifically suitable for publication and will be formally accepted for publication once it meets all outstanding technical requirements.

Kind regards,

Belgin Sever, Ph.D.

Academic Editor

PLOS ONE

Additional Editor Comments (optional):

Reviewers' comments:

Reviewer's Responses to Questions

**Comments to the Author**

1. Does the manuscript provide a valid rationale for the proposed study, with clearly identified and justified research questions?

Reviewer #1: Yes

2. Is the protocol technically sound and planned in a manner that will lead to a meaningful outcome and allow testing the stated hypotheses?

Reviewer #1: Yes

3. Is the methodology feasible and described in sufficient detail to allow the work to be replicable?

Reviewer #1: Yes

4. Have the authors described where all data underlying the findings will be made available when the study is complete?

Reviewer #1: Yes

5. Is the manuscript presented in an intelligible fashion and written in standard English?

Reviewer #1: Yes

You may also provide optional suggestions and comments to authors that they might find helpful in planning their study.

Reviewer #1: The authors have adequately addressed all of my earlier comments, so I have no further comments to add.

**Do you want your identity to be public for this peer review?** For information about this choice, including consent withdrawal, please see our Privacy Policy

Reviewer #1: No

---

## [Editor Report · Acceptance letter]

PONE-D-24-55750R1

PLOS ONE

Dear Dr. Nakamura,

I'm pleased to inform you that your manuscript has been deemed suitable for publication in PLOS ONE. Congratulations! Your manuscript is now being handed over to our production team.

Kind regards,

on behalf of

Assoc. Prof. Dr. Belgin Sever

Academic Editor

PLOS ONE